# Explaining the Gap Between the Ambitious Goals and Practical Reality of Animal Welfare Law Enforcement: A Review of the Enforcement Gap in Australia

**DOI:** 10.3390/ani10030482

**Published:** 2020-03-13

**Authors:** Rochelle Morton, Michelle L. Hebart, Alexandra L. Whittaker

**Affiliations:** School of Animal and Veterinary Sciences, The University of Adelaide, Roseworthy Campus, Roseworthy 5371, Australia; michelle.hebart@adelaide.edu.au (M.L.H.); alexandra.whittaker@adelaide.edu.au (A.L.W.)

**Keywords:** animal welfare legislation, animal cruelty, law enforcement, Australia, enforcement gap

## Abstract

**Simple Summary:**

Animal cruelty or neglect is an emotive social issue. Animal welfare legislation is the primary tool for defining, penalizing and hopefully deterring animal cruelty. A number of issues arising at all stages of the animal law enforcement process have been previously identified. These issues contribute to a discrepancy between the written law and the realities of the animal law enforcement process: the ‘enforcement gap’. This paper reviews the available resources to identify the causes of this ‘gap’. It is argued that the ‘gap’ is caused by numerous factors derived from all stages of the enforcement process: (1) reporting acts of animal cruelty, (2) ambiguity and shortcomings derived from the language used in animal welfare legislation (3) the nature of enforcement authorities, and (4) court determination on the matter. In order to reduce the enforcement gap and bring the expectations closer to reality, further research is needed.

**Abstract:**

Previous research has identified a number of issues arising at all stages of the animal law enforcement process. These issues contribute to an enforcement gap between the written law, as it relates to the penalties laid out in statutes, and the reality of the animal law justice system. This paper identifies and investigates the contributors to this gap. The identified factors discussed are (1) the role of the public in reporting animal cruelty, (2) the ambiguity of the language used in animal welfare legislation, (3) the nature of enforcement authorities, and (4) the role of the courts. Thus, the causes of the enforcement gap are multifactorial, derived from all stages of the enforcement process. Further research on the enforcement model and public education, in addition to debate on legislative reforms, will be needed to address this gap.

## 1. Introduction

Animal cruelty is a multiplex issue that affects animals globally [1]. In the last two decades, there has been an increase in public concern regarding issues of animal welfare [2,3]. With this growing concern, researchers have started to question the role and efficacy of the law in the regulation and promotion of animal welfare [2,4,5,6,7,8,9,10,11]. These authors have identified several weaknesses in the animal protection legal process in the Australian context, from the ambiguity of the language used in legislation [7] and the unorthodox use of non-government organizations (NGO) for enforcement of this branch of criminal law [7,9,12], to the severity of the penalties imposed for offences [8,10]. These weaknesses create a gap between the ambitious goals of animal law enforcement and the practical reality of the animal law justice system. Such a gap has been previously acknowledged and defined as an ‘enforcement gap’ in environmental law [13,14]. Lo, Fryxell, van Rooij, Wang and Li [13] defined this gap as a disparity between the practices laid out in the regulations and the actual practices of the regulations. To put it simply, there is an identified gap between the intentions or goals of written law and the outcomes of the enforcement process, as the goals are not meeting the expected outcomes. This ‘enforcement gap’ has not been defined or investigated in the animal law context and may be an issue experienced in many common law countries. However, this review will focus on Australian animal law. 

The ‘enforcement gap’ is a phenomenon that seeks to explain the reason why many well-intended laws fail [14], in terms of the expected outcomes not meeting the realities of enforcement. Qualifying the ‘expected outcomes’ of animal law enforcement can be challenging due to the subjectivity surrounding peoples ‘expectations. However, arguably the most objective way to qualify the ‘expectations’ is through understanding the intentions of the legislature at the time of bill drafting. Legislative intent refers to the interpretation and understanding of the reason behind the legislature’s enaction [15]. This can be done by analyzing the objectives as laid out in the legislation. In the case of animal protection legislation, this objective is to prevent animal cruelty by promoting animal welfare [16,17,18,19,20,21,22,23]. Another way to understand the scope of legislative intent is by examining the maximum penalties for offences, as the maximums provide a benchmark against which the gravity of an offence should be measured [24]. Maximum penalties have been the subject of several legislative reforms in Australia cross-jurisdictionally, with the states of Queensland (QLD) increasing maximum penalties in 2001 [25], South Australia (SA) in 2008 [26], Victoria in 2012 [27] and Northern Territory (NT) proposing to increase penalties in 2020 [28]. These amendments indicate the intention of Parliament to “get tough” on animal welfare offenders, by sending a message that animal cruelty will not be tolerated [29,30]. However, studies have shown that this movement to “get tough” on animal abusers is not being reflected by the Magistrates’ Courts in sentencing [9,10,24,31], as less than 10% of the maximum penalties are being used in court [10]. This implies, based on this definition, that legislative intent is not being achieved.

Public sentiment has been the catalyst for legislative reform. This was noted from statements made during consultations on the bill proposing the penal increases in SA. The Honorable Russell Wortley stated that “the proposed changes to this bill reflect the public’s concerns” [32]. However, sociological studies have indicated that the public regard the current penalties as too lenient [2,6]. As well as a desire for the criminal justice system to take crimes against animals more seriously [2,33], the Australian public are largely in favor of harsher penalties, such as imprisonment [2], for deliberate acts of animal cruelty. Whereas, in reality, terms of imprisonment are rarely handed down for animal welfare offences [9,10]. This poor use of statutory maximums, especially after they were the subject of substantial increases, arguably provides evidence that the enforcement of animal protection legislation is failing to meet the legislatures’ intentions, and, as a result, fails to act on the ‘public’s concern’. This creates the enforcement gap, as there is a discrepancy between the expectations of animal law enforcement and the current reality of the criminal justice system.

This paper seeks, through a literature analysis, to investigate this enforcement gap concept further, both identifying and describing the causes of the gap. The entire process, starting with reporting cruelty and ending with sentencing in court, as depictured in Figure 1, will be discussed. Theorized causes to be discussed include: (1) the inconsistencies and ambiguity of language used in animal welfare legislation, (2) the impact the public can have on reporting animal cruelty, (3) the reliance on NGO’s as enforcement bodies, and their unusual relationship with the Australian State and Territory governments and (4) the role of the courts. This review will discuss these topics, with a primary focus on the SA jurisdiction, with comparisons drawn to other Australian, and overseas states and territories. Throughout this article, the term ‘animal law’ is used to refer to statutes with the object of animal protection by the prevention of cruelty and promotion of welfare. 

## 2. Animal Welfare Legislation

### 2.1. Criminal Law

Animal law is a branch of criminal law. It serves to regulate the human treatment of, and behavior towards, animals in different contexts [12]. The law achieves this by outlining the basic duty of care that humans are legally required to provide to animals, and specifying what constitutes an offence against animals, through either commission of cruelty or a breach of the duty of care [12]. Offences are tried at the Magistrates Courts [12]. As criminal law, the prosecution must establish the elements of the case to a high standard of proof, that of ‘beyond reasonable doubt’, to secure a guilty finding. Generally, all animals are protected under the legislation, whether they are kept as pets, used for profit or declared as a pest [12].

There is no single overarching federal animal protection legislation in Australia, since there is no ‘head of power’ for animal welfare in the *Commonwealth of Australia Constitution Act* (‘Constitution’) [34]. Thus, animal welfare is a residual power, within the domain of the Australian states and territories [34]. Consequently, there are eight pieces of animal welfare legislation at the state and territory level in Australia (Table 1), hereafter termed ‘the Acts’. 

### 2.2. Inconsistency Issues

Animal welfare legislation differs between each state and territory in Australia [35,36,37,38,39,40,41,42]. The definition of an ‘animal’ is a key element in animal protection legislation. It is defined in each Act in a generally consistent manner. The term ‘animal’ is often any member of the vertebrate species, excluding human beings [12]. However, there is an inconsistency between the Acts. In the SA and Western Australian (WA) Acts, the definition of ‘animal’ does not include fish, whereas all other state and territory legislation includes fish as protected species. Another inconsistency is the type of offence. Some states and territories have aggravated cruelty offences, where intention (or recklessness) needs to be demonstrated as an element of the offence; the *mens rea*. This is in addition to duty of care offences, requiring persons to provide basic standards of care to animals. However, QLD, Tasmania and NT only include duty of care breaches [12]. Since the aggravated cruelty offences attract increased penalties, this leads to discrepancies between the maximum penalties for offences. Additionally, some states, such as New South Wales (NSW), use ‘penalty units’ to calculate monetary fines [36], whereas SA has set dollar amounts for the maximum monetary fines [39]. 

### 2.3. Problematic Definitions

Focusing on the SA *Animal Welfare Act 1985*, there are some issues surrounding specific terms and definitions in the Act. ‘Harm’, by definition, is “any form of damage, pain, suffering or distress (including unconsciousness), whether arising from injury, disease or any other condition” [43]. The issue then is the need for harm to have occurred before any legal action can be taken. An example of a provision where harm is required to be shown is in section 13(3)(b)(iv) of the *Animal Welfare Act 2985* (SA) below:

(3) Without limiting the generality of subsection (1) or (2), a person ill treats an animal if the person— 

   (b) being the owner of the animal— 

     (iv) neglects the animal so as to *cause it harm*; [44]

The fact that a person’s actions must “cause it harm” means that harm has had to happen and must be proven beyond reasonable doubt in court to secure a finding of guilt. This prevents the enforcement agency from intervening in potential animal abuse situations, as harm must occur to constitute an offence. If harm has not occurred yet, or there is dispute amongst experts as to the presence of harm, or insufficient evidence of that harm, then prosecution cannot be pursued. Therefore, it could be argued that the *Animal Welfare Act 1985 (SA)*, does not allow for the prevention of cruelty, it only allows the prosecution thereof. This appears to be in direct contradiction with the objective of the Act [16], which is the *promotion* of animal welfare. 

The WA legislature does not have this issue, as the *Animal Welfare Act 2002* includes the words “likely to cause harm” [45] instead of a singular “harm”. An example of the application of harm from Section 19(3) of the amended *Animal Welfare Act 2002* (WA) is below:

(3) Without limiting subsection (1) a person in charge of an animal is cruel to an animal if the animal — 

   (a) is transported in a way that causes, *or is likely to cause*, it unnecessary harm; or

   (b) is confined, restrained or caught in a manner that —

     (i) is prescribed; or

     (ii) causes, *or is likely to cause*, it unnecessary harm; [45]

This allows the enforcement agencies in WA to intervene before harm has occurred, thus protecting animals by promoting their welfare. According to an update on the WA Government website, the Department of Agriculture and Food have successfully prosecuted a person transporting livestock in a way that was likely to cause harm [46]. This provides some measure of success associated with the new wording. However, the effect of the amendment on prosecution rates has not been reported. Accordingly, it may be too early to draw firm conclusions on both the impact on animals, and any consequent increased evidential burden on the enforcement agency and difficulties meeting this burden. 

The discussed issues in Australian animal welfare legislation require further investigation and consideration. However, it is hypothesized that they contribute to the enforcement gap by preventing the enforcement agencies from carrying out their enforcement responsibilities to the full extent of the law. 

## 3. Animal Law Enforcement and the Public

### 3.1. Non-Government Organizations

Generally, individual government agencies, such as state police forces, are given power under the general criminal law to enforce legislation. In the case of animal law in Australia, an NGO, namely the Royal Society for the Prevention of Cruelty to Animals (RSPCA), has the bulk of the enforcement burden. This is achieved through the inspectorate division [7,47,48]. These inspectors are given power of enforcement after being appointed by the Minister of each state or territory’s relevant government department [35,36,37,39,40]. The relationship between the RSPCA, and Australian state/territory governments has an extensive history, with the SA government entrusting the RSPCA SA to enforce animal welfare laws for more than 100 years [49]. This relationship is a reflection of our colonial past [47]. The English RSPCA began enforcing anti-cruelty laws from its inception (as the SPCA), since a police force was not in existence at that time [47]. It is important to note that the RSPCA in Australia is a federated organization, with member societies in each state and territory, and RSPCA Australia as the national body [50]. RSPCA Australia has no role in enforcing the state and territory animal welfare laws. Instead RSPCA Australia has a role in influencing animal welfare policy, practice and legislation [50]. The enforcement is the responsibility of the state and territory RSPCAs, as independent organizations. 

The state and territory-based RSPCAs are involved in animal law enforcement in each state and territory of Australia, with the exception of NT, where a government department (Department of Primary Industries and Resources) enforces the legislation entirely [51]. The enforcement burden is often shared between two agencies in each state and territory (Table 2), through the signing of a memorandum of understanding [48,52,53]. This shared agreement often sees the RSPCA enforcing companion animal matters and government-run departments enforcing livestock cases [48,52,53,54,55]. 

### 3.2. Reporting Cruelty

The first step of the enforcement process is reliant on the public. This step involves contacting one of the above enforcement agencies (Table 2) to report the details of an act of animal cruelty. Reporting cruelty as a whole contains several variables that come into play. From previous research it is understood that the propensity to report criminal acts (not exclusively involving animals) involves the following factors: (1) presence or absence of witnesses, (2) financial losses incurred, (3) the seriousness of the crime [65,66], (4) distrust in the relevant enforcement agency, (5) and fear of retaliation [67]. Although these studies are not predominantly animal-crime-orientated, they can be used as a guide to the decision-making process someone undertakes before reporting animal cruelty. An individual’s likeliness to report animal abuse was further examined by Taylor and Signal [68]. This study had the major findings that (1) people working within the livestock industry had the lowest propensity to report animal abuse and (2) although people have the intention to report, they do not know whom to report to. 

The first major finding of Taylor and Signal [68], that people who work with livestock are less likely to make animal cruelty complaints across all species, was discussed by Kellert [69], where this lower propensity was related to the strong utilitarian views these workers have. That is, they show greater concern for the animal’s practical and material value, rather than solely their welfare. This, however, does not imply that those working within the livestock industry condone or engage in acts of animal abuse, they just have a different attitude towards animals, influencing their propensity to report [70], which can be related to their experiences in working within the livestock industry [71]. A further consideration of demographics may also be important. Individuals in rural populations, in comparison to urban populations, may be less likely to report cruelty because livestock ownership is comparatively more ‘hidden’ than companion animal ownership, especially in the case of intensive industries such as piggeries and poultry farms. It is theorized that, because of the often secluded and geographically dispersed nature of the livestock industry, evidence of abuse is harder to obtain, the cases are expensive to prosecute because of the number of animals involved, and it has been suggested that prosecutors often view them as risky [72]. Furthermore, recent debate or enactment of “ag-gag” laws comes in to play. These laws aim to hinder or ‘gag’ discussions on more controversial facets of the animal agricultural sector through curbing animal activist and investigative activities [73]. They prevent public oversight and criticism of livestock practices and have the potential to allow animal cruelty to remain unseen, thus reducing the number of cruelty reports. Although these laws are more prevalent in the US [74], they have been emerging in Australia, as discussed by Englezos [73].

The second major finding of the Taylor and Signal [68] study was that a large proportion of people (27% of their sample) were unaware of how to report acts of cruelty, leading them to either not report, or report to agencies not tasked with the bulk of animal law enforcement. This is a major issue and has recently been validated again in Australia by Glanville, et al. [75], where a survey of the Victorian public found that the majority of participants would not report to the primary enforcement authority (RSCPA Victoria). In fact, a significant number of responses (27%) indicated that the public would take no action if they witnessed animal mistreatment. After investigation into the reasons behind inaction, the authors found that 24% of respondents were uncertain whether animal mistreatment was actually taking place. However, when asked to assess their confidence levels in recognizing animal mistreatment, the majority of participants rated themselves with high confidence. Therefore, it appears that perceived knowledge does not always equate to action. 

There is further evidence of public uncertainty as to what constitutes an animal welfare offence. RSPCA Victoria have stated that the community expectations of animal welfare are much higher than the legal standard the RSCPA has to work within [76], implying that the public misunderstand what constitutes an animal welfare offence, and consequently misreport to the RSPCA. Sophie Buchanan, Head of Prevention at RSPCA Victoria, stated:

“One of the things that we continually observe is that community understanding of what constitutes an offence under the Prevention of Cruelty to Animals Act and the reality of the act are quite separate. Community expectations about welfare are quite high, but the threshold for an offence under the act is quite significant. Even though these are summary offences, the level of harm that has to be proven that an animal has suffered makes the threshold for investigation and prosecution quite significant. So rather than vexatious, the majority of unsubstantiated complaints or reports that we receive relate more to people’s misunderstanding of what would constitute an offence”[76]

Thus, even though there is some evidence that the public either do not report animal abuse, or that they report to inappropriate enforcement authorities [68,75], there is still an issue with people reporting acts which do not meet the legal criteria for an offence. Rectification of reporting issues is likely best achieved through public education, focused on the identity of the appropriate enforcement agency and what constitutes an animal welfare offence. Relaying to the public the need for the enforcement agency to establish the elements of the statutory provision, as well as the evidential burden for prosecution, is necessary in advancing public understanding. If the public do not understand this concept, they may become frustrated with the enforcement system. Education of this nature is something the state and territory RSPCAs have begun implementing on their websites, see e.g., [49]. This will allow the enforcement agencies and the public to work towards a relationship that is filled with understanding, and consequently will reduce the numbers of misreports and general underreporting of offences. 

## 4. Animal Law Enforcement Agencies

### 4.1. Conflicts of Interest

Animal law enforcement is carried out by two different types of enforcement authorities: government-run departments, and NGOs. Government departments in each state and territory, other than the Australian Capital Territory (ACT) and SA, are tasked with farm animal welfare enforcement [77]. This regulatory relationship between government-run agricultural departments and animal welfare enforcement has challenges, specifically the conflict of interest between promoting the livestock industry economically and also regulating it to serve public interest [77]. Sociological research has indicated that public concern regarding issues of farm animal welfare is growing [78,79,80], shifting from the traditional utilitarian philosophy to a more empathic approach [81,82,83]. Consequently, the public expect advancements in farm animal welfare to match their growing expectations. However, it has been suggested that the Agricultural Departments in Australia fail to balance promoting the industry for profit and protecting farm animal welfare to meet public interest [77]. Goodfellow [77] has discussed this topic in depth, arguing that good welfare practices and on-farm productivity do not come hand-in-hand, despite the information from industries suggesting otherwise, see, e.g., [84]. The main point of discourse is that protecting animal welfare is not always the most profitable option, leading to a conflict for the Departments in choosing between the promotion of the industry or protecting animal welfare. Goodfellow [77] argued that the former is the obvious choice for the industry, given previous patterns in history, and concluded that reforms are required to rectify this imbalance. 

NGO enforcement is undertaken primarily by Australian state and territory RSPCAs, which are also involved in advocacy campaigns for animal-related issues, as per the organizations’ objective “to educate the community with regard to the humane treatment of animals” [85]. However, conflicts of interest arise when advocacy campaigns imitate activism activity [86]. According to the Oxford Dictionary, the definition of advocacy is “public support for, or recommendation of, a particular cause or policy” [87], while activism means “the policy or action of using vigorous campaigning to bring about political or social change” [88]. Although both words refer to creating a change through means of public expression, the differing factor is the way the expression is conducted: activism seems to take a more forceful approach. In the Victorian report on the RSPCA inspectorate division, Comrie [86] discussed the public’s difficulty in differentiating between the RSPCA’s activism work and law enforcement responsibilities, arguing that there is a conflict of interest since the law fails to align with the RSPCA’s core beliefs and values. However, the overarching objective of animal welfare statutes is to prevent animal cruelty by promoting animal welfare [16,17,18,19,20,21,22,23], and the RSPCA Australia’s mission is “to prevent cruelty to animals by actively promoting their care and protection” [89]. The RSPCA promotes the concept of ‘welfare’ as maintaining the ‘five freedoms’ of animals [90], and it has been suggested that animal welfare legislation in Australia is underpinned by the ‘five freedoms’ [91]. This indicates that there is no conflict of interest between the legislative objectives and the RSPCA’s core values. 

Despite this apparent alignment between legislative objectives and RSPCA’s values, at face value there could be a conflict of interest between the activism work in which RSPCA is involved, and lawful activities as authorized by animal welfare legislation. An example of this was discussed by the Victorian Legislative Assembly Committee of Economy and Infrastructure on RSPCA Victoria’s fitness as an enforcement agency [92]. Representatives of the Sporting Shooters Association of Australia raised concerns about the RSPCA’s stance on hunting as a recreational activity [92]. This is a legal activity in Victoria under the Code of Practice for the Welfare of Animals in Hunting (revision no. 1) [93]. The issue of contention was that RSPCA Victoria was campaigning against duck hunting, as per RSPCA Australia’s policy of being

“opposed to the hunting of any animal for sport as it causes unnecessary injury, pain, suffering, distress or death to the animals involved”[94]

However, given the legality of duck hunting in Victoria, RSPCA Victoria were campaigning against lawful activity. This same issue has been noted in campaigns on dairy cows, greyhound racing, layer hens, live exports, meat chickens, pig farming and whips in horse races [95], which are all legal activities under Victorian law [92], and common campaigns run by RSPCAs across Australia. This conflict between the RSPCA’s campaigning work and enforcement responsibility has been suggested to not only confuse the public, but also compromise the relationship between RSPCA Victoria, government officials, and members of hunting, sporting and primary production organizations, making them reluctant to engage with the NGO [86]. RSPCA Victoria have since acknowledged this issue and released a response to the Comrie review [86], stating that the organization will “focus on achieving improvements in animal welfare by using trust-based advocacy approaches” [96]. RSPCA Victoria voiced that all public campaigns will be focused exclusively on direct owner care of animals, not advocating against Victorian laws [76]. RSPCA Victoria or any other state/territory NGO enforcement authorities have had no reported issues with animal welfare campaigns since the review. 

In an independent review on animal welfare enforcement in WA, it was recommended that the inspectorate divisions within both NGO and Government Departments (Department of Agriculture and Food WA), were kept separate from the operational areas, to avoid any potential conflicts of interest [97]. Thus, since providing NGO enforcement and campaigning activities work independently from one another as two completely separate entities, the risks of conflict should be diminished.

### 4.2. Resourcing Issues

Each financial year, state governments fund the individual RSPCAs to cover the costs of enforcement. However, according to the annual reports available on the RSCPA SA’s website, the government funding they received for the 2018/2019 financial year only covered 43% of the total cost to enforce SA’s relevant animal welfare legislation (*Animal Welfare Act 1985* (SA)) [98], the remainder being funded at their own expense. There is no other branch of criminal law that relies so heavily on charity to ensure the enforcement of what is essentially a public interest law [12,99]. The position of RSPCA Victoria in regard to their enforcement responsibilities was well summarized by Magistrate D.J. Faram during a Victorian animal welfare case in 2015. His Honour said: 

“The RSPCA in particular is a statutory body with prosecutorial powers but without significant support from Parliament. They are also the body charged with rescuing and rehabilitating these animals. This comes at a significant cost for an organization that receives some state funding but otherwise relies on donations and bequests and other fundraising activities”[86]

In the 2018/2019 financial year, the South Australian inspectors responded to 4244 cruelty reports [98]. Of those reports, only 32 were prosecuted in court, meaning only 0.8% of the complaints investigated resulted in charges being laid. These low prosecution rates are not exclusive to South Australia, as the cumulative prosecution rate for Australia also equates to 0.5% (Table 3). 

This small percentage of prosecutions in relation to reports is assumed within the literature to be caused by a lack of resources available to the enforcement agencies, most commonly the state and territory-based RSPCAs [72,105]. The resourcing issue is often referred to in monetary terms [72]. However there is minimal evidence to support this claim, nor the ability to measure the magnitude of its effects; there is only speculation. 

Within the literature, it has been suggested that the RSPCA will only prosecute cases of a grievous nature, where evidence is overwhelming, as a way of saving resources [72,105]. That is, the organization will only prosecute cases where guilty verdicts are likely to be assured due to the risk of adverse cost orders in the event of a not guilty finding. If true, this position may preclude the bringing of test cases and some neglect cases where impact on animal wellbeing may be less easy to characterize. An example of these test cases could include cases of mental suffering experienced by animals, which is not widely recognized in animal law [11], despite the scientific evidence in support of animals experiencing emotion [106,107,108,109]. As an example, the issue of obese pets is becoming increasingly prevalent [110] and has been legally tested in 2007 in the UK, see, e.g., [111]. Two brothers were found guilty of causing unnecessary suffering to their dog by allowing the dog to become so obese that he was “effectively crippled” [111]. Cases of this nature appear not to have been tried in Australia. The doctrine of precedent allows the law to develop through case law, since precedents in certain circumstances are legally binding, meaning that judges must follow the determinations and rulings of judges in higher courts. However, if test cases are not initiated to test the boundaries of legal terminologies such as animal “harm” or “suffering”, the only opportunity for animal law to progress is through statutory reform [112]. The latter requires a groundswell of support, and is retrospective and often lengthy [112]. These test cases have significant value not only for incrementally progressing animal law, but also in guiding statutory interpretation and safeguarding animal welfare. Therefore, the speculated under-resourcing has the potential to impact on animal law development through impeding the progression of the doctrine of precedent. 

Nonetheless, prosecution is only one way to promote animal welfare and it may not be an optimal method. In the WA independent review of animal welfare enforcement, both the RSPCA and the Department of Agriculture and Food WA, who share the enforcement responsibilities, made a submission to the review based on their belief that education was the most appropriate method of achieving compliance with the legislation [97]. Education can either be in the form of widespread community outreach, through campaigns or seminars, or legally sanctioned education in the form of animal welfare notices/directions. Animal welfare notices are essentially ‘instructions’ given to owners to rectify any duty of care breaches before prosecutorial action is taken. These notices are outlined under the state and territory Acts [113,114,115,116,117,118,119,120] and are given to owners by animal welfare inspectors or officers. It is an offence if the owners fail to comply with the notices, and they can also become further evidence of an offence, to be used later in court. 

This suggests that enforcement agencies will use prosecution as a tool only when the education of an individual has failed. Thus, they will only prosecute cases if necessary, as a last resort, not because they are required to prioritize the cases of the worst nature due to resourcing concerns. However, it is possible that the cases of the worst nature are in fact the cases where education would fail, as these animal abusers are often displaying levels of moral numbness [8], rendering education ineffective, and punitive measures more appropriate. It is probable, in reality, that decision-making as pertains to course of action does not follow a clear step-by-step ‘trial’ of increasingly more punitive measures, instead being more nuanced, based on individual case facts and personal experiences of enforcement personnel. Another point of note, given that both the RSPCA, as a charitable organization, and the Department of Agriculture and Food, as a government department, both gave preference to education over prosecution when they have access to different levels of resourcing, is indicative that the low prosecution rate is not necessarily reflective of resourcing strains experienced by the RSPCA. This focus on education goes back to the RSPCA’s mission to prevent cruelty by promoting animal welfare, which is where their name came from: Royal Society for the *Prevention* of Cruelty to Animals [85], not the *‘Prosecution’* of Cruelty to Animals. 

There is, however, evidence in support of a resourcing issue: in statements made during the consultation on the bill proposing changes to the *Animal Welfare Act 1985 (SA)*, the Honourable Mark Parnell stated:

“What is the point of increasing penalties if we do not increase the resources that are used to investigate cases of animal cruelty?” [32]

Further evidence in support of this is from the independent review of animal welfare enforcement in both Victoria [86] and WA [97]. Comrie [86] acknowledged the resourcing issue by suggesting potential alternative resourcing strategies. The main suggestions were to delegate some of the tasks of the inspectorate division to other departments within the NGO, engage more volunteers in inspectorate functions, and work more closely with local government agencies for a greater widespread animal welfare network. Since the Comrie review [86], it has been stated that RSPCA Victoria has made significant improvements in this regard [95], however, the nature of the improvements were not commented on. Thus, no conclusions can be drawn on whether the apparent resourcing issue has improved for RSPCA Victoria. In the WA review, Easton, Warbey, Mezzatesta and Mercy [97] acknowledged the need to allocate additional resources to animal welfare, however the authors noted that they did not have access to any statistical data, and derived their conclusions based on the submissions made to the review, which may have been influenced by bias. 

The statement made by RSPCA Victoria cited earlier, referencing the public misunderstanding of what constitutes cruelty under law [76], is further evidence disputing the resourcing issue. It can be argued that the alleged resourcing issue, as measured by the substantial gap between reports and prosecutions, is due to the public misreporting acts of animal welfare concerns, rather than an inability to fully investigate due to inadequate resourcing. In order to reduce this issue of misreporting, RSPCA have run campaigns to better educate the public on animal cruelty investigations; an example is RSPCA SA’s ‘combat cruelty’ campaign, see, e.g., [121]. It is likely that both inadequate fund provisions to enforcement agencies, as well as public misunderstanding of what constitutes an animal welfare offence play a part in the low prosecution rates presented in Table 3. However, without the guidance of any statistical support or claims from the enforcement agencies themselves, the existence or extent of a resourcing issue is mere conjecture. 

### 4.3. Alternative Enforcement Models

Other countries have implemented different methods for animal law enforcement. The Auckland SPCA of New Zealand do not receive any government funding for their enforcement work [122]. This organization relies on fundraising and donations to fund their investigation work. However, they have formed a pro bono panel of lawyers who offer their time and experience to prosecute animal welfare offences on behalf of the Auckland SPCA [122]. In Killeen’s review [122] of this pro bono panel in 2013, it was reported that the panel have been very successful in securing severe penalties for offences and arguing for legislative reform. An article on New Zealand’s Law Society website confirms that this pro bono panel was still in action in March 2017, and comprises approximately 40 lawyers [123]. Recently, the Canadian province of Ontario has removed the SPCA from their enforcement role all together, after concerns over charitable dollars subsidizing a government function [33]. Coulter [33] suggested that Ontario should consider a strategic combination of a government division for enforcement and not-for-profits for support and animal care. This partnership model has been successfully implemented in New York City, where the police force and ASPCA now work symbiotically, with police officers trained for animal cruelty investigations and the ASPCA providing support [33]. 

In Australia, from the two independent reviews of animal welfare enforcement in Victoria [86] and WA [97], and the Victorian Government’s inquiry into RSPCA’s fitness as an enforcement agency [95], the final conclusions drawn are that the current model in place, where NGOs enforce animal welfare statutes, is the most appropriate. The assumption that transferring enforcement responsibility to the government will lead to greater transparency may be too simplistic [5]. Currently, in jurisdictions where government agencies are tasked with animal law enforcement, commonly involving cases of farm animal abuse [48,52,53,54,55], little to no information on their enforcement activities is made publicly available [55]. In contrast, the RSPCA release figures each financial year detailing their enforcement activities [98,101,102,103,104,124]. 

## 5. The Court Process

### 5.1. Court Level

Animal welfare offences are generally initiated in the Magistrates Court in Australia [12]. Being a lower court, any decisions made are not binding to future cases, and are unreported, resulting in a lack of precedents. Decisions are only binding when made in higher courts. This usually only applies to animal law appellate cases presented to the higher courts. As discussed earlier, law develops incrementally through case law [112]. The absence of precedents renders animal law to be somewhat of a ‘statutory beast’. 

### 5.2. Maximum Penalties

Penalties for animal welfare offences were the subject of numerous amendments in SA [10], QLD [31] and Victoria [62], where the maximum penalties for offences were increased. Although the maximum penalties for offences are significant, for example, in SA the maximum custodial sentence is 4 years and the monetary fine is $50,000, on average, less than 10% of these maximums are being used in court [10]. This raises questions regarding the purpose of maximum penalties. It is suggested that they are reserved for the worst, most serious examples of an offence [125], but if they are never applied, are they just a symbolic gesture? It has been suggested that terms of imprisonment and fines are not the most effective way to punish animal abusers, and alternative penalties should be considered [8,10]. Such reasoning is derived from the fact that imprisonment as a penalty for criminal offences meets very few of the punishment theory facets, being deterrence, rehabilitation, retribution, restitution and incapacitation [126,127,128,129]. When considering animal law, imprisonment only completely satisfies the retribution aspect, as the suffering experienced by the animal harmed is compensated by the suffering experienced by the offender in jail [127]. Other facets of punishment theory seem to be ignored or are not fully satisfied. Alternative penalties, such as court-mandated counselling, should aim to reduce the prevalence of reoffending and prevent the likelihood of later human violence, as there is an established link between animal abuse and human violence [130,131,132,133,134,135,136,137]. However, as noted by Holoyda [138], it is unclear what type of counselling should be imposed, since there are no models for the treatment of animal abuse that have undergone any sort of study or peer review. Despite uncertainty regarding the effectiveness of alternatives penalties, their consideration is still warranted to prevent further acts of cruelty, whether to animals or humans. 

### 5.3. Prohibition Orders

Enforcement agencies commonly apply to the court for prohibition orders as a mechanism to further protect animals [139], as stated by RSPCA QLD’s Prosecutions Officer, Tracey Jackson:

“In many cases, the best way to protect animals, is to use prohibition orders to limit or regulate animals owned by offenders. Many times people offend due to their current circumstances, and they simply need a break from having animals, or from having so many animals”.[139]

A prohibition or supervision order is a direction issued by the court to an offender found guilty of an offence, the provisions of which are outlined in the state and territory Acts [140,141,142,143,144,145,146,147]. These orders either prevent offenders from owning animals for a set period or until a further order is approved by the courts, or limit the number of animals to be owned [140,141,142,143,144,145,146,147]. Supervision orders work similarly to prohibition orders. The difference is that the offender is not prohibited from owning animals; they are under the supervision of an enforcement agency, who can regularly check on the welfare of the offender’s animals [148]. 

The issue with these court orders is that they are not cross-jurisdictionally recognized, allowing people who have been prosecuted and placed under a prohibition order for animal welfare offences in one state to cross the border and be free of the provisions therein. In a recent scenario, dog breeders pleaded guilty to animal welfare offences in Victoria and were placed under a nine-year prohibition order from owning animals [149]. The breeders entered SA and continued to breed dogs, since the Victorian prohibition order no longer had effect. The breeders have since been alleged to have committed offences under the *Animal Welfare Act 1985* (SA) and have had animals seized from them [150]. They are currently undergoing prosecution in the SA Magistrates’ Court [149]. In an opinion piece to the South Australian Law Society, the Chief Executive Officer for RSPCA SA, argued for the recognition of interstate prohibition orders, as it would discourage people with prohibition orders from relocating to another state and continuing the ownership of animals [151]. Currently, in Australia, the recognition of interstate orders has been achieved for domestic violence orders [152] and is undergoing debate by members of the QLD Parliament for firearm prohibition orders [153]. Thus, there is a legal precedent for making such a change in relation to animal welfare offences.

### 5.4. Case Sentencing

Currently fines, generally considered as the least severe option, are the most common penalty given, while imprisonment, being the most severe option, is the least common [9,10,24,154]. This is in contradiction to public expectations, since they appear to largely favor prison sentences [2,155]. However, case sentencing is a complex process and, as Markham [24] noted, analysis of sentencing data would be a difficult task. There is no strict formula for sentencing cases; there is human (judicial) input, as well as any factors that the court will take into account. For this reason, the remaining discussion on case sentencing does not proclaim to be a comprehensive analysis of the entirety of the process, nor does it intend to undermine the current processes in place or underestimate the complexities of the criminal justice system. This discussion is simply a review of the available literature in relation to case sentencing and the ways in which it may contribute to the enforcement gap concept. 

In court, the judge will use the maximum penalty as a guide and decide the sentence they see fit in accordance with the relevant sentencing legislation, taking into account aggravating and mitigating factors. In SA, the sentencing legislation in place is the *Sentencing Act 2017* [156]. This piece of legislation provides guidance to the courts when sentencing offenders for criminal offences [157]. For example, section 10 of the *Sentencing Act 2017* outlines the use of imprisonment sentences, stating: 

(2) Subject to this Act or any other Act, a court must not impose a sentence of imprisonment on a defendant unless the court decides that— 

   (a) the seriousness of the offence is such that the only penalty that can be justified is imprisonment; or

   (b) it is required for the purpose of protecting the safety of the community (whether as individuals or in general) [158]

This specifies that, in SA, terms of imprisonment must only be handed out as a last resort, which is in contradiction to public expectations, which largely support prison sentences for animal welfare offences [2,155]. Public support for prison sentences is not unique to animal law; general community opinions on sentencing in criminal law are that courts are ‘too lenient’ and ‘not severe enough’ [159]. This suggests that the public are not aware of the complexity of case sentencing, and that terms of imprisonment cannot be handed out for any case. Judges have a legal obligation to not misuse prison sentences for a myriad of reasons, one being to not utilize costly prison resources on offenders who do not pose a risk to the community, as costs associated with incarcerating an individual in Australia are high [160]. Misuse of those resources would result in over-stretched prison systems, which would prevent the incarceration of criminal offenders who pose a serious risk to the community. Considering that the majority of animal abuse cases are negligence cases, where the offender has failed to provide an animal with adequate living conditions or veterinary treatment, not aggravated offence cases, where the offender intended to harm the animal [10], this implies that the majority of animal abusers do not pose a risk to the community. Therefore, other rehabilitative forms of penalty are likely to be more appropriate and cost-efficient. Despite the fact that the public are largely in favor of prison sentences, they may not necessarily be warranted for the majority of perpetrators of animal crimes. 

Animal law in Australia is enforced by various organizations, including the police, and their case outcome data, and the nature of case files is assumed to be individualized for each organization. Evidence for this can be seen from the review of animal cruelty offences in Victoria, where the authors, McGorrery and Bathy [62], acknowledged six different organizations/departments that supplied the data for their analysis. The lack of any centralized tracking of case information challenges the ability to access and analyze any sentencing trends, operate effectively across agencies, provide recommendations and make use of the doctrine of precedent. It has been reported that the US FBI have begun tracking animal abuse cases [161], which is an idea worth consideration in Australia. 

## 6. Conclusions

The enforcement gap in animal law is created as a result of a range of factors derived from all stages of the enforcement process, from cruelty reports to sentencing. The causes of the enforcement gap are not exclusive to those mentioned in this review and this concept is certainly not isolated to animal law, but this area of law has some unique elements due to the unusual reliance on a charitable organization for policing. There is a dearth of empirical data on contributors to this gap, and further research is needed on the following concepts; (1) the differences between the Australian state/territory animal welfare legislation, (2) education programs to better inform the public on the means of reporting animal cruelty (3) the extent of the resourcing issues experienced by NGOs, and (4) alternative penalties for offences. Without such evidence, it is challenging to suggest solutions to the problem, but it is likely that a combination of changes to the enforcement model, legislative reform and public education is required. 

## Figures and Tables

**Figure 1 animals-10-00482-f001:**
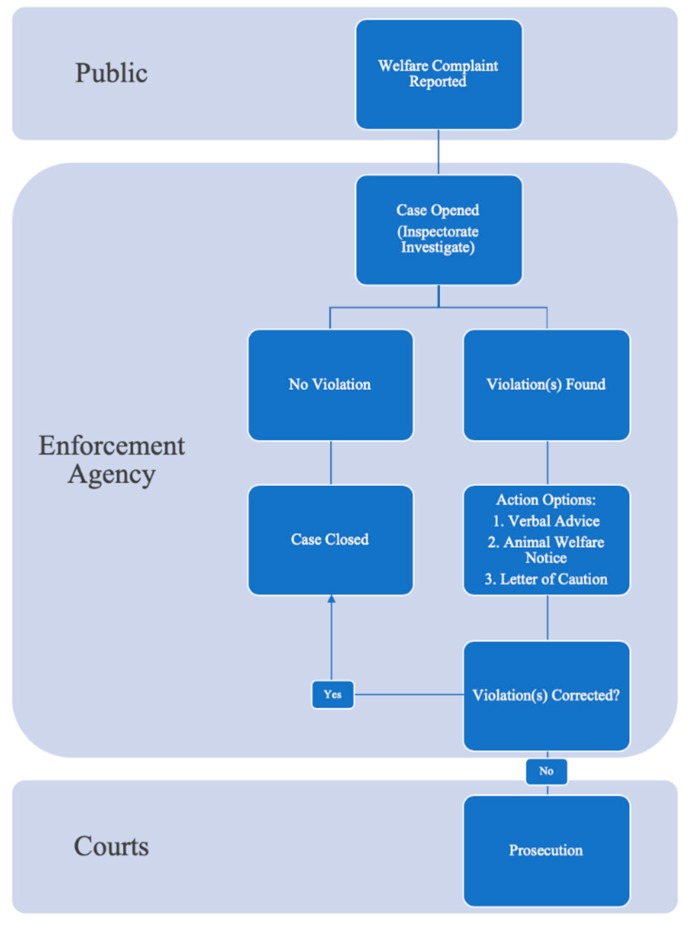
Flowchart of the Australian animal law enforcement process, indicating the parts where the public, enforcement agencies or the courts have the predominant power.

**Table 1 animals-10-00482-t001:** Each state and territory’s relevant animal welfare legislation in Australia.

State/Territory	Animal Welfare Legislation
Australian Capital Territory	*Animal Welfare Act 1992* [35]
New South Wales	*Prevention of Cruelty to Animals Act 1979* [36]
Northern Territory	*Animal Welfare Act 1999* [37]
Queensland	*Animal Care and Protection Act 2001* [38]
South Australia	*Animal Welfare Act 1985* [39]
Tasmania	*Animal Welfare Act 1993* [40]
Victoria	*Prevention of Cruelty to Animals Act 1986* [41]
Western Australia	*Animal Welfare Act 2002* [42]

**Table 2 animals-10-00482-t002:** Each state and territories’ relevant animal welfare legislation and predominant enforcement agencies in Australia.

State/Territory	Enforcement Agency ^1^
Australian Capital Territory	RSPCA ^2^ Australian Capital Territory [56]
New South Wales	RSPCA NSW [57] Animal Welfare League NSW [57]
Northern Territory	Department of Primary Industries and Resources [51]
Queensland	RSPCA Queensland [58] Biosecurity Queensland [59]
South Australia	RSPCA SA [49]
Tasmania	RSPCA Tasmania [60] Department of Primary Industries, Parks, Water and Environment [61]
Victoria	RSPCA Victoria [52] Department of Job, Precincts and Regions [52] Department of Environment, Land, Water and Planning [62]
Western Australia	RSPCA WA [63] Livestock Compliance Unit [64]

^1^ All State and Territory police forces are given powers to enforce the animal welfare legislation. ^2^ RSPCA refers to the Royal Society for the Prevention of Cruelty to Animals.

**Table 3 animals-10-00482-t003:** Numbers of inspectors, cruelty reports and prosecutions for each state and territory in Australia. All data were gathered from the 2018/2019 RSPCA annual reports. Note, as described in Table 2, that the RSPCA may not be the sole enforcement body for that state or territory. The Northern Territory is not included in Table as RSPCA NT has no role in enforcement.

State/Territory	RSPCA Inspectors	Cruelty Reports	Prosecutions	Prosecution Rate (%)
Australian Capital Territory	3 [86]	1024 [100]	1 [100]	0.1
New South Wales	32 [101]	15,673 [101]	77 [101]	0.5
Queensland	24 [86]	17,810 [102]	154 [102]	0.9
South Australia	9 [98]	4244 [98]	32 [98]	0.8
Tasmania	4 [60]	1900 [100]	10 [100]	0.5
Victoria	26 [103]	11,638 [100]	94 [100]	0.8
Western Australia	15 [86]	6417 [104]	10 [104]	0.2

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
