# Peer review of "Explaining the Gap Between the Ambitious Goals and Practical Reality of Animal Welfare Law Enforcement: A Review of the Enforcement Gap in Australia"

_animals, 2020, doi:10.3390/ani10030482_

Round 1

Reviewer 1 Report

Valuable work, but needs to improved.

Overall comment
This is an interesting and important topic. However, while the authors have done a good job in places in this paper, they have not quite got across several issues. I strongly recommend they spend some time talking to RSPCA Australia, notably Dr Jed Goodfellow, who has unique insight into this area. Furthermore, many of the references cited are somewhat dated and in some cases irrelevant. This needs tightened up.

However, with a bit more work, this should be a very relevant and useful contribution.

General structure
I think the paper needs to be reorganized, with the section on legislation at the beginning. This then paves the way to introducing the role of the RSPCAs, as each different jurisdiction, with different laws, has different RSPCAs involved in enforcement. I think it would also be helpful to have a historical comment on why the RSPCA ‘model’ has been adopted. It should not be forgotten that at the time of Richard Martin’s Act in England, there was no police force as such. Enforcement and prosecution were matters for private citizens. Hence the first SPCA was a ‘prosecution society’.

The paper needs to set out the status of each of the States and Territories RSPCAs and the existence and role of RSPCA Australia. The important point, of course, is that the States and Territories RSPCAs are independent organisations, each with their own constitutions and governing body. As the authors point out, the relevant legislation is also unique to each jurisdiction. RSPCA Australia has a role in coordination and policy setting. It also has an important role in providing high level guidance and reporting on animal welfare science.

I think it will also help greatly if the authors can point out at a very early stage that animal ‘cruelty’ law is criminal law, which means that proof of an offence has to be at a high level – beyond reasonable doubt. This has important implications regarding likelihood of success of prosecutions, and may not be understood by many outside the arena.

The title of the paper indicates the authors are focused on the ‘enforcement gap’ in Australia. References to the situation in Australia are in the main not very current. There is also a tendency to make reference to non-Australian work when making statements about the Australian situation, or statements in general. I think in this regard that references 18, 26 and 104 should be removed, as they concern the USA, the UK and New Zealand.

References to sections in statutes do not need page numbers as well.

The authors should reference RSPCA Australia sources regarding statistics, eg https://www.rspca.org.au/sites/default/files/RSPCA%20Australia%20Annual%20Statistics%202017-2018.pdf.

The paragraph referring to appointment of officers of ‘charitable organisations’ under the various animal welfare (etc) acts is confused. The key point is that the local RSPCAs have either no role in their appointment (eg as is the case in Tasmania) or can nominate a person for approval, or an employee can be approved.

References to the Northern Territory in the first para of 2.2 should be deleted. The correct position is stated in the second para.

The comment at line 101 regarding the distinction between enforcement relating to companion animals versus livestock needs a bit more research. My view is that the livestock community have very negative views of the RSPCAs, preferring that the ‘agriculture’ departments are involved in enforcement. You can get a feel for that (as an example) in the Upper House parliamentary inquiry in WA which happened a few years ago. The reference to Tasmania in Table 1 is incorrect.

I suggest the authors will get a better picture of what is going on here by talking to Dr Jed Goodfellow at RSPCA Australia.

An important point which is missed is the role of issuing ‘instructions’ regarding action needed to be taken to alleviate animal welfare problems. This in my view contributes to the apparent low ratio of prosecutions to complaints, and is relevant in the education context.

The statement about Canada at line 387 is wrong. The action referred to was in Ontario.

As I read the section concerning the roles of RSPCAs in the different jurisdictions, and especially the part where the authors refer to the majority of complaints relating to companion animals (line 102), I felt uncomfortable that there was a lack of deep analysis relating to what goes on with livestock (in particular). In my view, the majority of cruelty to animals in Australia is inflicted on livestock. To some extent this is masked by the inclusion of exemptions and ‘defences’ in various parts of the relevant legislation, but equally it is in my view arguable that a lot of livestock-related cruelty is masked more by two factors. One is that much of it (and in particular intensive livestock ‘farming’) is out of sight of the public (and the authors deal with this well), and the other is that the enforcement responsibility is with departments who are probably more concerned with being the ‘farmers’ friend’ than enforcement. The subject of ‘regulatory capture’ in that regard is covered in depth in Dr Jed Goodfellow’s PhD thesis (Macquarie University).

The comment that livestock prosecutions are risky can be illustrated by the 2015 Vic RSPCA case involving many cattle which were euthanased, an action found by relevant courts to be unnecessary (see RSPCA v Holdsworth [2015] VSCA 243).

The reference to Commonwealth law at 407 is irrelevant. This paragraph should start with the statement that animal welfare legislation is the responsibility of States and Territories, as there is no appropriate head of power in the Commonwealth Constitution. – so move this from line 429.

However, the rest of the section on the legal status of animals, and the definition of ‘animal’ is largely irrelevant to the subject of the paper. Likewise the section on Uniform Acts. All of this should be deleted.

Author Response

This is an interesting and important topic. However, while the authors have done a good job in places in this paper, they have not quite got across several issues. I strongly recommend they spend some time talking to RSPCA Australia, notably Dr Jed Goodfellow, who has unique insight into this area. Furthermore, many of the references cited are somewhat dated and in some cases irrelevant. This needs tightened up.

However, with a bit more work, this should be a very relevant and useful contribution.

Dr Jed Goodfellow is actually an advisor for the corresponding author's PhD. Since this review submission he has the document and provided  feedback. Thank you for this suggestion.

I think the paper needs to be reorganized, with the section on legislation at the beginning. This then paves the way to introducing the role of the RSPCAs, as each different jurisdiction, with different laws, has different RSPCAs involved in enforcement.

We do agree with this point. The legislation section is now after the introduction, which is then followed by the enforcement sections and the sentencing court section.

I think it would also be helpful to have a historical comment on why the RSPCA ‘model’ has been adopted. It should not be forgotten that at the time of Richard Martin’s Act in England, there was no police force as such. Enforcement and prosecution were matters for private citizens. Hence the first SPCA was a ‘prosecution society’.

This is a good point, we have added the section on the history of the RSPCA on lines 167-172.

The paper needs to set out the status of each of the States and Territories RSPCAs and the existence and role of RSPCA Australia. The important point, of course, is that the States and Territories RSPCAs are independent organisations, each with their own constitutions and governing body. As the authors point out, the relevant legislation is also unique to each jurisdiction. RSPCA Australia has a role in coordination and policy setting. It also has an important role in providing high level guidance and reporting on animal welfare science.

We have added a brief discussion on RSPCA Australia on this on lines 172-176.  

I think it will also help greatly if the authors can point out at a very early stage that animal ‘cruelty’ law is criminal law, which means that proof of an offence has to be at a high level – beyond reasonable doubt. This has important implications regarding likelihood of success of prosecutions, and may not be understood by many outside the arena.

Another good point, we have added a brief introduction on criminal law on lines 85-91.

The title of the paper indicates the authors are focused on the ‘enforcement gap’ in Australia. References to the situation in Australia are in the main not very current. There is also a tendency to make reference to non-Australian work when making statements about the Australian situation, or statements in general. I think in this regard that references 18, 26 and 104 should be removed, as they concern the USA, the UK and New Zealand.

References 18 and 26 have been removed. However, we have opted to keep reference 104 (now reference 111) included on line 478, as it is stating that animals can experience emotion and the author of that reference David Mellor from NZ is a pioneer in that area of research. In terms of outdated  references, we have tried to use the most recent information relevant to this area of research. Having checked them again, we are confident that all thereferences are still relevant and add merit to thereview.

References to sections in statutes do not need page numbers as well.

All page numbers have been removed.

The authors should reference RSPCA Australia sources regarding statistics, eg https://www.rspca.org.au/sites/default/files/RSPCA%20Australia%20Annual%20Statistics%202017-2018.pdf.

The statistics from the state and territory RSPCA were used as there were slight discrepancies between the States’ data and those from RSPCA Australia’s annual statistics. We opted to use the state/territory statistics as we believed these would  be more accurate.

The paragraph referring to appointment of officers of ‘charitable organisations’ under the various animal welfare (etc) acts is confused. The key point is that the local RSPCAs have either no role in their appointment (eg as is the case in Tasmania) or can nominate a person for approval, or an employee can be approved.

After consideration we understand how this section is confusing. We have  included the section saying the RSPCA can be appointed by a Minister on line 166. The rest of the section has been   removed to avoid confusion.

References to the Northern Territory in the first para of 2.2 should be deleted. The correct position is stated in the second para.

Thank you for picking that up. The incorrect reference (41) has now been deleted.

The comment at line 101 regarding the distinction between enforcement relating to companion animals versus livestock needs a bit more research. My view is that the livestock community have very negative views of the RSPCAs, preferring that the ‘agriculture’ departments are involved in enforcement. You can get a feel for that (as an example) in the Upper House parliamentary inquiry in WA which happened a few years ago.

We have opted to remove this line from the paper as we want the focus of the paper to be on enforcement in general, not focusing on specific groups of animals. We do agree with you and would love to include an in-depth discussion comparing the two enforcement models, but do not think the space is available to do justice to that in this review. 

The reference to Tasmania in Table 1 is incorrect.

We apologise but are  unsure if this comment refers to formatting of the reference or the government carrying out enforcement. We have referenced the government website saying that they carry out livestock inspections, however have kept the RSPCA Tas reference in as it states that they carry out enforcement in addition. The formatting error of that reference has also been fixed

I suggest the authors will get a better picture of what is going on here by talking to Dr Jed Goodfellow at RSPCA Australia.

We do agree that Jed would provide a good insight into this topic, for that exact reason Jed is an advisor for the PhD work in this area, and he regularly reviews the work and makes suggestions for improvements. In regard to the livestock versus companion animals’ discussion, although we agree with your comments, we have tried not to include specific discussions on livestock animal or companion animals apart from brief discussion on the memorandum of understandings between RSPCA’s and state/territory governments. The aim of this review was to focus on enforcement in general, not delve too deep into how enforcement differs between each type of animal. It is a very valuable discussion though which is not lost on us

An important point which is missed is the role of issuing ‘instructions’ regarding action needed to be taken to alleviate animal welfare problems. This in my view contributes to the apparent low ratio of prosecutions to complaints, and is relevant in the education context.

Another great point, referring to animal welfare notices. We have included a brief discussion on this on line 496.

The statement about Canada at line 387 is wrong. The action referred to was in Ontario.

Thank you for picking this up, the error has been fixed (refer to line 561). 

As I read the section concerning the roles of RSPCAs in the different jurisdictions, and especially the part where the authors refer to the majority of complaints relating to companion animals (line 102), I felt uncomfortable that there was a lack of deep analysis relating to what goes on with livestock (in particular). In my view, the majority of cruelty to animals in Australia is inflicted on livestock. To some extent this is masked by the inclusion of exemptions and ‘defences’ in various parts of the relevant legislation, but equally it is in my view arguable that a lot of livestock-related cruelty is masked more by two factors. One is that much of it (and in particular intensive livestock ‘farming’) is out of sight of the public (and the authors deal with this well), and the other is that the enforcement responsibility is with departments who are probably more concerned with being the ‘farmers’ friend’ than enforcement. The subject of ‘regulatory capture’ in that regard is covered in depth in Dr Jed Goodfellow’s PhD thesis (Macquarie University).

We understand this comment but feel that it is difficult to give this discussion the depth it deserves in this review. Our aim was to try and keep the discussion of speciesism (companion animal vs livestock) to a minimum.  We have included a section on government enforcement of livestock welfare on lines 412-432, since this is a valuable discussion point. Hopefully this will show the readers that there is a lot more to this issue.

The comment that livestock prosecutions are risky can be illustrated by the 2015 Vic RSPCA case involving many cattle which were euthanased, an action found by relevant courts to be unnecessary (see RSPCA v Holdsworth [2015] VSCA 243).

This is a very interesting case, thank you for bringing it to our attention. We have opted not to include it in this review, as this is a tricky case to analyse due to the unlawful euthanasia. Although it does demonstrate how livestock cases can be risky, we also think RSPCA Vic probably did not handle the case well at that point in time (2003) which is likely a one-off incidence rather than a systemic issue in the enforcement model.

The reference to Commonwealth law at 407 is irrelevant. This paragraph should start with the statement that animal welfare legislation is the responsibility of States and Territories, as there is no appropriate head of power in the Commonwealth Constitution. – so move this from line 429.

This has been remedied and edited as per this recommendation.

However, the rest of the section on the legal status of animals, and the definition of ‘animal’ is largely irrelevant to the subject of the paper. Likewise the section on Uniform Acts. All of this should be deleted.

These sections have been removed.

Reviewer 2 Report

This paper is trying to show enforcement gap in animal welfare but does not define this gap so it is impossible to assess if the gap exists or how it will be assessed.  The paper tries to assess weaknesses in the enforcement model by looking at several stages of enforcement - however as the definition of what the enforcement gap is not mentioned it is impossible to assess each of these stages to assess if there is an enforcement gap.  The stages are all valid stages (ie reporting cruelty, investigating authority and reliance on a ngo to do this work, law itself, sentencing) but the the paper does not defined or assess these stages objectively but appears to be focused on the role of the RSPCA in this area rather than objectively assessing each stage.  If the title of the paper was the role of a ngo in enforcing animal welfare legislation it would be a fairer assessment of the paper - the paper appears to start with the position of saying there is no role for a nog in enforcing animal welfare law but agrees it does not have the data to do this objectively and so relies on recent inquiries into the RSPCA in Victoria and West Australia to do so rather than objectively looking at the data and what the public and Government want to see.  

The paper suffers from a major error - by not setting out what the goal of animal law enforcement should be it cannot assess if there is an enforcement gap.  I agree the goal of enforcement is difficult eg is it more or less prosecutions, more or less complaints on cruelty, harsher or softer sentencing but as there is no objective goal agreed the paper cannot meet its goal of assessing if a gap exists (it seems from the paper that a gap does not exist but this is opposite from what the paper assesses).  The paper then does not assess each of the four stages to see which has a gap and which has the greatest gap so is flawed as it cannot make any conclusions on where the gap should be rectified and how.

The paper suffers from a number of factual errors and its references are negatively biased towards the role of the RSPCA.  Underneath are a few examples

l101 the reason why the RSPCA is tasked with non farm animal issues is four fold: 1. the government are tasked with farm animals as they are an agency priority as they are a food item 2. the public expects the RSPCA to do domestic animal enforcement (as measured by polling) 3. there is a potential conflict between RSPCA enforcing farm animals and RSPCA Assured scheme so this conflict is removed by not enforcing those laws 4. RSPCA has experience in dealing with domestic animals whereas the government and vets have experience in dealing with farm animals  l142-5 this is a dangerous assumption that the public do not phone RSPCA due to a distrust and is not backed up by the statistics (eg phone calls, public trusts in RSPCA) - indeed no references are given to weigh up this statement which is patently untrue as the public do trust RSPCA with cruelty complaints more than the Government and there is a greater public desire to see the RSPCA undertake this work than the Government (measured via public polling and branding work) l 181 this is untrue - there is no confusion on what happens to the cruetly complaint - RSPCA Victoria website gives detailed information on issues, actions and what happens to cruelty complaints l 196 these data are incorrect - the latest RSPCA SA report shows an income of $1.1 million from fundraising and not 3 million costs and 7.2 million to fund animal operations l 199 this is incorrect the RSPCA Victoria report shows 30% not 50% spent on animal operations division - the Government fund $1.1 million of this - it is unclear why erroneous figures are used but it seems to be to prove the argument rather than objectively assessing the facts l207 this fact (the need to re-establish public confidence) is not referenced and is not borne out by brand and polling work  l211 there is no mention of the seperation by RSPCA of enforcement, prosecutions and campaigning work which is crucial for them to undertake these different tasks  l224 the paper then says there is an apparent alignment between RSPCA values and legislative objectives but this appears to be contrary to what has been argued in lines 188-211 l 238 there is no mention if RSPCA Victoria carry out enforcement work on the issues raised - dairy cows, greyhounds, horse racing but I believe they do not so there is no conflict of interest lines 243-255 there is a lot of cherry picking from the  Cowrie Report the report also states that it has a "favourably impressed with the commitment and professionalism of the inspectorate" and that "no other organisation is as well equipped as the RSPCA to deal with animal cruelty and no other organisation could fill the breach".  It is concerning that Cowrie has been cherry picked so I can only assume that this is to validate the argument that RSPCA cannot enforce the law rather than objectively assess the facts which would show otherwise l269 again this is cherry picking from the UK - the EFRA inquiry into the animal enforcement in the UK and the work of the RSPCA found "The RSPCA has an invaluable role in investigating allegations of animal mistreatment. We recognise that the organisations fulfills a role in animal welfare not currently performed by local government" and the Government stated that "Government does not consider, at this time, that the RSPCA should be made a specialist reporting authority. Instead we believe that the RSPCA should be given the opportunity to implement the recommendations of the Wooler Review and demonstrate its commitment to responding to the concerns that have been raised by the Committee."  I wonder at the authors' intent to cherry pick on the UK and Australia inquiries.  l 276 the separation of inspectorate and operational areas are indeed separate in RSPCA in UK and Victoria but the paper does not reference this  ls 282-9  the argument here does not make sense ie the Government part funds the RSPCA but the charity money on enforcement means there is a conflict - this is not necessarily so.  The RSPCA is funded by the public and enforcement and investigation is one of the top motivators for donations - this is not referenced but would suggest that the public want the RSPCA to undertake this work line 295-303 this is very muddled and erroneous - partly this is because the authors have not decided what is the goal of animal welfare enforcement - is it to reduce prosecutions or increase, to reduce cruelty complaints of increase.  This lack of a goal for the paper is at the heart of why this section is factually incorrect.   l 303 there is no evidence to show the small percentage of prosecutions is due to a lack of money - indeed the opposite is the case.  Prosecutions are only taken if they meet the DPP tests in Australia and UK, hence the high success rate.  If RSPCA took more prosecutions they would rightly be accused of taking erroneous malicious prosecutions.  Indeed the paper in line 306 agrees there is no evidence to support this allegation so it is unclear why it is mentioned line 309 this is incorrect - as above the two DPP tests must be met before a prosecution is undertaken l 314 this is wrong - the reason why mental suffering is not prosecuted more is the difficulty of meeting the DPP tests line 317 this case is ironically a RSPCA UK case from 2007 - so the authors are using a single RSPCA case from the UK to argue that more prosecutions should be taken on obesity whilst also saying the RSPCA should not be prosecuting as they have no Government and public support - all these assumptions are not only incorrect they undermine the argument the paper is trying to construct l 326 this is incorrect as above and there is no evidence to support this line nor do the authors put any reference together  l377 the authors are contradicting themselves here  line 387 this is factually incorrect - Canada is run on a State basis not a Federal one and the Ontario SPCA removed themselves from enforcing the law not the Federal Government so all this section is inaccurate line 402 no reference to polling which exists for the RSPCA - there is ample evidence to show the public have a high regard for the RSPCA enforcement and pay money through donations for it to enforce the laws  lines 617-9 this is the section that says the paper is irrelevant - there is some empirical evidence on the gap but the authors do not reference it nor establish what the gap is.  The conclusion suggests that the paper does not meet its goals - this referee would agree with this assessment

Author Response

This paper is trying to show enforcement gap in animal welfare but does not define this gap so it is impossible to assess if the gap exists or how it will be assessed.  The paper tries to assess weaknesses in the enforcement model by looking at several stages of enforcement - however as the definition of what the enforcement gap is not mentioned it is impossible to assess each of these stages to assess if there is an enforcement gap. 

The enforcement gap is defined as weaknesses in animal welfare enforcement, which creates this gap between the goals of enforcement and the current reality. This definition is outlined on lines 43-45.

The stages are all valid stages (ie reporting cruelty, investigating authority and reliance on a ngo to do this work, law itself, sentencing) but the the paper does not defined or assess these stages objectively but appears to be focused on the role of the RSPCA in this area rather than objectively assessing each stage.  If the title of the paper was the role of a ngo in enforcing animal welfare legislation it would be a fairer assessment of the paper

When focusing on animal welfare enforcement in Australia, it is impossible not to discuss the roles of RSPCAs in this model. RSPCAs were only discussed in two section of this review (Animal Law Enforcement and the Public and Enforcement Agencies). The sections relating to the legislation and the sentencing process include no discussion on the RSPCAs. We therefore assert that the paper title is appropriate, being focused on animal law enforcement, which the RSPCA happens to be a part of.

The paper appears to start with the position of saying there is no role for a nog in enforcing animal welfare law but agrees it does not have the data to do this objectively and so relies on recent inquiries into the RSPCA in Victoria and West Australia to do so rather than objectively looking at the data and what the public and Government want to see.  

We apologise but cannot pinpoint the line in the paper that implies there is no role for ngo’s in animal law enforcement. In regard to not objectively looking at data that the public and Governments want – the inquires used were Government-issued inquiries and thus would be assumed to provide an idea of what the state/territory governments are hoping to achieve from animal law enforcement. The WA inquiry was also focussed mainly on the Government department responsible for enforcement rather than that RSPCA. In terms of the viewpoint of the general public, there is not a lot of available literature on this, hence the research gap referred to. One of the most recent studies was Taylor & Signals 2009 study ‘Look ‘em up and Throw Away the Key? Community Opinions Regarding Current Animal Abuse Penalties’, which I referred to several times. If I have missed some peer reviewed studies regarding these topics, I apologise and would appreciate if you could refer them on to me.

The paper suffers from a major error - by not setting out what the goal of animal law enforcement should be it cannot assess if there is an enforcement gap.  I agree the goal of enforcement is difficult eg is it more or less prosecutions, more or less complaints on cruelty, harsher or softer sentencing but as there is no objective goal agreed the paper cannot meet its goal of assessing if a gap exists (it seems from the paper that a gap does not exist but this is opposite from what the paper assesses). 

The objective of this paper was not to assess the goal of animal law enforcement as that goal is extremely subjective. If anything, the goal is outlined in the state and territory Acts, which is to put simply ‘to promote welfare and prevent cruelty’, the means of meeting those goals is up to the enforcement agencies and the sentencing courts. As mentioned in lines 47-50, the enforcement gap is caused by anything that impedes the ‘expected’ outcome of animal law enforcement. From statements made from enforcement agencies (from the inquiries), animal law academics (the literature) and the public (social studies), it is apparent that animal law enforcement is not meeting it goals as per legislative intent and this is what creates the gap. The enforcement gap concept is not unique to animal law and is used in the legal literature, hence we applied the definition used by legal scholars.

The paper then does not assess each of the four stages to see which has a gap and which has the greatest gap so is flawed as it cannot make any conclusions on where the gap should be rectified and how.

We believe each of the four stages have been discussed with reference to the available literature under a relevant subheading. The intent was not to try and quantify a gap numerically- since the ‘gap’ is a theoretical and subjective notion. The conclusions made in each section were that without further empirical research these gaps are merely speculative, and the concepts require further investigation and analysis before construing them as fact. For example, in lines 789-792, it is concluded that without evidence it is challenging to suggest solutions to the problems, however it is likely that a combination of structural changes to enforcement agencies, legislative reforms and public education is required. It is extremely difficult to make conclusions on the issues raised in this paper as there is little empirical data to perform objective analyses, hence the need for further research.

The paper suffers from a number of factual errors and its references are negatively biased towards the role of the RSPCA.  Underneath are a few examples

l101 the reason why the RSPCA is tasked with non farm animal issues is four fold: 1. the government are tasked with farm animals as they are an agency priority as they are a food item 2. the public expects the RSPCA to do domestic animal enforcement (as measured by polling) 3. there is a potential conflict between RSPCA enforcing farm animals and RSPCA Assured scheme so this conflict is removed by not enforcing those laws 4. RSPCA has experience in dealing with domestic animals whereas the government and vets have experience in dealing with farm animals 

Thank you for providing this information. We have  opted to remove this statement as it was not intended to discuss the differences between companion animal enforcement and livestock enforcement, but focus on animal law enforcement more generally.

l142-5 this is a dangerous assumption that the public do not phone RSPCA due to a distrust and is not backed up by the statistics (eg phone calls, public trusts in RSPCA) - indeed no references are given to weigh up this statement which is patently untrue as the public do trust RSPCA with cruelty complaints more than the Government and there is a greater public desire to see the RSPCA undertake this work than the Government (measured via public polling and branding work)

This information is not presented as a fact but an author suggestion to be unpacked in the review. Therefore a reference is not needed. We have included a statement that it has not been investigated on line 255 to ensure the reader does not misconstrue the information as a fact. We are unable to find any peer reviewed literature on this topic. We apologise if important references have been missed.

181 this is untrue - there is no confusion on what happens to the cruetly complaint - RSPCA Victoria website gives detailed information on issues, actions and what happens to cruelty complaints

We respectfully disagree with this comment. Just because the information is available on the website does not mean that the public have read and understand what happens to cruelty complaints. Without questioning of the public on their awareness, it is challenging to either prove or dispute this claim. In the text we have included the word ‘may’, to indicate that this information is not a fact, it is an opinion. We have included a statement saying that this information is available on RSPCAs’ websites to inform the readers on line 292.

196 these data are incorrect - the latest RSPCA SA report shows an income of $1.1 million from fundraising and not 3 million costs and 7.2 million to fund animal operations

The data presented was on the expenses/costs not the income. Personally, I have triple checked these numbers and had another reviewer check these numbers for accuracy.

According to the 2017/2018 annual statistics the costs are as below:

Enforcing Animal Welfare Act : $2,795,949

Animal Operations: $5,189,661

Fundraising/Marketing: $3,339,714

You can find this information from reference 71 in the paper or alternatively the link is here: https://www.rspcasa.org.au/wp-content/uploads/2018/09/RSPCA_South-Australia_Annual-Report_2017-18.pdf

I used 2017/2018 data because at the time of writing this review this information was the only information available, and the monetary values are not included in this much detail on the most recent 2018/2019 SA statistics since reporting categories appear to have been changed.

199 this is incorrect the RSPCA Victoria report shows 30% not 50% spent on animal operations division - the Government fund $1.1 million of this - it is unclear why erroneous figures are used but it seems to be to prove the argument rather than objectively assessing the facts

The 50% referred to related to the above figures of RSPCA SA, it had nothing to do with RSPCA Victoria. Due to this confusion, we have included “SA” in line 324, so the readers do not confuse the sentences.

207 this fact (the need to re-establish public confidence) is not referenced and is not borne out by brand and polling work 

This is not presented as a fact; it is presented as an opinion and recommendation for future peer reviewed research. As the statements made about RSPCA Victoria may have been negatively influenced by the controversy they were involved in, we believe qualitative research in this realm would be very valuable. Also, it was regarding public opinion, not confidence.

211 there is no mention of the seperation by RSPCA of enforcement, prosecutions and campaigning work which is crucial for them to undertake these different tasks 

This statement was made to lead into the next section on conflicts of interest. We state that the enforcement work is separated from advocacy work in lines 408-410.

224 the paper then says there is an apparent alignment between RSPCA values and legislative objectives but this appears to be contrary to what has been argued in lines 188-211

In lines 188-211 (now 298-336) (section 4.1) the relationship between RSPCA, as an enforcement agency, and the Australian public, as a point of cruelty reports is being discussed. Legislative objectives were not mentioned in this section. The section relates to the potential for the public to become confused by all the different activities the RSPCA is involved in (enforcement, advocacy, education, animal operations, fundraising). It is our opinion that the public may become confused and that there may be conflicts of interest where an organisation is campaigning for legislative change when they are also enforcing within that same legislative framework. However, line 224 (now 349), refers to the fact that although the core values of RSPCA (prevent cruelty by promoting welfare) align with the objectives of animal welfare acts (promote welfare and prevent cruelty), there still may be conflicts of interest present within RSPCA internally. To simplify it, we are saying that the RSPCA can enforce animal laws as their values align with the objectives of the Acts, however, there still may be conflicts of interest between RSPCA advocating for legislative change and enforcing the same legislation. To soften the wording, we have replaced “does appear” with “could” on line 350.

238 there is no mention if RSPCA Victoria carry out enforcement work on the issues raised - dairy cows, greyhounds, horse racing but I believe they do not so there is no conflict of interest lines

According to the Memorandum of Understanding between RSPCA Victoria and the Department of Jobs, Precincts and Regions, RSPCA Victoria’s Inspectorate are responsible for all complaints against RSPCA Animals as outlined in Schedule 1 (below). This includes small hobby dairy farms (less than ten cows), greyhounds in racing and horses in racing. This  information can be found here: http://agriculture.vic.gov.au/__data/assets/pdf_file/0004/483970/MoU_RSPCA_Victoria_and_DJPR_2019-2024.pdf

Schedule 1 – RSPCA Animals

For the purpose of clause 3.1 of this MoU, ‘RSPCA Animals’ means:

(a) companion and recreational animals;

(b) primary production animals where less than ten (10), which includes cattle, sheep, pigs, goats, deer and fifty (50) in the case of poultry;

(c) equids, including horses used in riding schools and in standardbred or thoroughbred racing; and

(d) greyhounds used for greyhound racing.

243-255 there is a lot of cherry picking from the  Cowrie Report the report also states that it has a "favourably impressed with the commitment and professionalism of the inspectorate" and that "no other organisation is as well equipped as the RSPCA to deal with animal cruelty and no other organisation could fill the breach".  It is concerning that Cowrie has been cherry picked so I can only assume that this is to validate the argument that RSPCA cannot enforce the law rather than objectively assess the facts which would show otherwise

In lines 243-255 (now 373-385) the Comrie Report is not mentioned.  We  assume the reviewer is referring to line 257-266 (now 387-396). The comments referring to cherry picking information make us uncomfortable- since we tried to present a balanced, critical argument based on available evidence.  The information used relates to the topic of that section, ie  the conflicts of interest being  discussed. The statements the reviewer has provided have nothing to do with conflicts of interest and thus seem irrelevant to the section.  The argument being presented is not about RSPCAs’ inability to enforce animal laws, it is about assessing the current model for any potential weaknesses that could create this gap between the expectations and reality of animal law enforcement. We are reviewing the available literature on animal law enforcement, which heavily involves RSPCA. We also make a similar conclusion to the quotes provided by the reviewer in lines 568-581, where it is stated that the current model in place where RSPCA enforces law is the most appropriate and it is too simple to think that the Government will do any different or better.

We have also included a discussion on government conflict, as we understand the entire section was focused on RSPCA and that is not fair. Please refer to lines 412-438 to read the government conflicts of interest.

269 again this is cherry picking from the UK - the EFRA inquiry into the animal enforcement in the UK and the work of the RSPCA found "The RSPCA has an invaluable role in investigating allegations of animal mistreatment. We recognise that the organisations fulfills a role in animal welfare not currently performed by local government" and the Government stated that "Government does not consider, at this time, that the RSPCA should be made a specialist reporting authority. Instead we believe that the RSPCA should be given the opportunity to implement the recommendations of the Wooler Review and demonstrate its commitment to responding to the concerns that have been raised by the Committee."  I wonder at the authors' intent to cherry pick on the UK and Australia inquiries. 

Again, we do not believe this information has been cherry-picked.  This section presents the available information on conflicts of interest between the RSPCA campaigning against laws they are required to enforce. Again, these quotes provided by the reviewer have nothing to do with the topic of this section ie conflicts of interest. We are not discussing whether or not the RSPCA should enforce laws in this section – we have this discussion in lines 568-581, where it is  concluded that the RSPCA is the most appropriate organisation to enforce animal laws, which is similar to the quotes provided by the reviewer.

276 the separation of inspectorate and operational areas are indeed separate in RSPCA in UK and Victoria but the paper does not reference this 

This is a separate paragraph and the literature being discussed relates to RSPCA WA, which is referenced in line 407 (reference no 93). RSPCA UK or Victoria have not been referenced in this paragraph since we do not have any references relating to them. It has also been concluded in line 408 (the next sentence) that because of this WA review, the readers can assume that all RSPCA keep their inspectorate separate from their operational areas. This appears to be reflect the sentiment expressed in this comment.

282-9  the argument here does not make sense ie the Government part funds the RSPCA but the charity money on enforcement means there is a conflict - this is not necessarily so.  The RSPCA is funded by the public and enforcement and investigation is one of the top motivators for donations - this is not referenced but would suggest that the public want the RSPCA to undertake this work line

This section is no longer about conflicts of interest. On line 281 (now 429) the subheading has changes to Resourcing Issues. This paragraph is simply providing the reader with some background about the RSPCAs resourcing model as per their annual statistics. Thus, there is no need to reference the public's motivation for donating since this is not being discussed.  Also, we apologise, but are unaware of any literature on the public’s motivation for donating to the RSPCA. If possible, could you refer us to these studies?

295-303 this is very muddled and erroneous - partly this is because the authors have not decided what is the goal of animal welfare enforcement - is it to reduce prosecutions or increase, to reduce cruelty complaints of increase.  This lack of a goal for the paper is at the heart of why this section is factually incorrect.  

Again, we apologise – but are unsure how the information provided is erroneous or factually incorrect. The information presented is available from the RSPCA annual statistics on their website as some background information on prosecutions. This leads into the following paragraph, which discusses how the literature seems to pair low prosecutions rates with poor resourcing. We do not believe prosecution rates are a way of measuring animal law enforcement goals, and do not believe this sections suggest this.

303 there is no evidence to show the small percentage of prosecutions is due to a lack of money - indeed the opposite is the case.  Prosecutions are only taken if they meet the DPP tests in Australia and UK, hence the high success rate.  If RSPCA took more prosecutions they would rightly be accused of taking erroneous malicious prosecutions.  Indeed the paper in line 306 agrees there is no evidence to support this allegation so it is unclear why it is mentioned

This information is not presented as a fact.  We wrote, “is assumed to be caused by a lack of resources” on line 303 (now 461). It is mentioned as this topic is the next point for the argument to be discussed.  It was the topic statement. In line 461, it is stated that the literature assumes or speculates that the low prosecution rates are caused by a lack of resources, and in line 464, it is stated that is this only speculation as there is no available evidence. From line 471, we then proceed to discuss this literature and present arguments for and against this resourcing issue as part of a critical analysis.  I hope this clears up why it was mentioned.

line 309 this is incorrect - as above the two DPP tests must be met before a prosecution is undertaken

This information is not presented as fact, it is stated that within the literature it is speculated that the RSPCA will only prosecute cases they know they’ll win as a way of saving resources. We then go on to discuss this issue and even say that prosecution is not the only way to promote animal welfare and talk about education from line 492.  

314 this is wrong - the reason why mental suffering is not prosecuted more is the difficulty of meeting the DPP tests

In line 314 (now 475) it is stated that mental suffering is not widely recognised in animal law, which the reviewer’s information that mental suffering does not meet the DPP tests confirms. If the tests struggle to recognise mental suffering, then that implies that the system fails to include cases of mental suffering, which causes mental suffering to not be recognised in animal law. It was never stated that a lack of funds causes mental suffering cases to not be recognised in animal law. It was stated that a lack of funds would make it hard to test the boundaries of law.

line 317 this case is ironically a RSPCA UK case from 2007 - so the authors are using a single RSPCA case from the UK to argue that more prosecutions should be taken on obesity whilst also saying the RSPCA should not be prosecuting as they have no Government and public support - all these assumptions are not only incorrect they undermine the argument the paper is trying to construct

We apologise for the confusion, but it was never stated nor implied that the RSPCA should not be prosecuting as they have no government or public support.  Actually the opposite is stated on lines 568-581, where it is concluded that this model (RSPCA enforcing laws) is the most appropriate model. We used the UK example as a way of showing the readers what a test case is, and there isn’t an example of one in Australia that is publicly available. It was to help explain the importance of the doctrine of precedent, not argue that more prosecutions should be taken on obesity. We were also not hiding the fact that it was from the UK and have included that is it from 2007 so the readers are more informed.

326 this is incorrect as above and there is no evidence to support this line nor do the authors put any reference together 

This line wasn’t presented as a fact which is why there is no reference. It was said that is was speculative and that is could potentially impact animal law development through the doctrine of precedent. To avoid any more confusion, we have included the words “doctrine of precedent” in line 489.

377 the authors are contradicting themselves here 

This is not a contradiction, it was a conclusion as it is the end of the discussion on the resourcing section. It was concluded that the resourcing issue is speculation, and that more evidence is required to affirm or deny whether there is an actual resourcing issue.

line 387 this is factually incorrect - Canada is run on a State basis not a Federal one and the Ontario SPCA removed themselves from enforcing the law not the Federal Government so all this section is inaccurate

Thank you for picking this up, we have changed ‘Canada’ to ‘Ontario’ on line 561.

line 402 no reference to polling which exists for the RSPCA - there is ample evidence to show the public have a high regard for the RSPCA enforcement and pay money through donations for it to enforce the laws 

Unfortunately, the RSPCAs polling does not include methodological information rendering it  difficult to critically analyse these surveys in order to conclude if they are a representative sample of the Australian population. For example, if the RSPCA was to distribute these surveys from their own databases, then this would create a bias and possibly not provide representative results of the Australian public. For this reason, these data have not been included in this review as part of a review of peer-reviewed literature.  

lines 617-9 this is the section that says the paper is irrelevant - there is some empirical evidence on the gap but the authors do not reference it nor establish what the gap is.  The conclusion suggests that the paper does not meet its goals - this referee would agree with this assessment 

In  our experience it is common when writing a literature review to identify the gaps in the research in order to inform future research.  We believe that a range of evidence has been presented arguing the existence of the gap (163 references). The goal of the paper was to suggest what the causes of the gap is, and we believe we have done this and analysed them critically, providing both sides of the argument.  We conclude that the enforcement gap is multifactorial (caused by the numerous stages of the enforcement process discussed in this review) but without further research we cannot suggest ways to rectify issues discussed.

Reviewer 3 Report

There are some other additional possible causes of the enforcement gap that are worthy of mention in addition to the 4 listed. One is the complexity of animal cruelty offenses that require different areas of expertise and professional orientation. Animal cruelty can take many forms from simple neglect, ‘organized” abuse such as dogfighting and cockfighting, harm to pets in the context of domestic violence, gang-related activity, commercial exploitation such as in ‘puppy mills’ or extreme neglect seen in cases of animal hoarding. Responding to this range of offenses requires collaborative efforts that go beyond the expertise of traditional law enforcement or animal protection and often require the resources of mental health, social services, victim advocates, forensic veterinarians and others. A holistic approach should involve efforts for collaborative task forces and opportunities for cross-training in these different disciplines.

A second additional barrier is the lack of well-defined criteria for recognizing the physical and behavioral evidence associated with animal abuse and neglect. This has only recently begun to be addressed in the emerging fields of veterinary forensics and forensic applied animal behavior. Given growing public expectations of forensic evidence in any prosecution, the ‘CSI effect’ of the absence of such evidence can be a barrier to charging or fully prosecuting a cruelty case. Although these fields are growing in the US, UK and Brazil - there has not been much attention to these fields in Australia.

A third additional contributor to the ‘gap’ is the absence of any centralized tracking of animal cruelty cases at the state/territorial or national level.  This has been a major issue in the US, with the FBI only beginning to attempt to track such cases in the last 3 years, with limited results so far.

lines 131-140 - In addition to the stated obstacles to prosecution of livestock cruelty cases, it may be worth mentioning that at least in the US there are specific livestock industry efforts to  directly block the investigation of such industries through hotly debated legislative efforts to pass “Ag-gag” laws that outlaw or severely restrict such investigations. I do not know if there have been such efforts in Australia, but they issue is worth mentioning.

lines 369-374  It is worth mentioning any public campaigns that serve to better educate the public to recognize and report suspected animal cruelty, including specific high profile crimes such as dogfighting. As noted, a key need is also to provide the public with up-to-date information on where to report suspected animal cruelty, In the US a good resource for finding appropriate responders is provided at nationallinkcoalition.org.

line 390 There is slight error here. The collaborative effort between the ASPCA and the New York Police Department involves 33,000 police officers. I believe the 14,000 number for Coulter’s study represents the number trained as of her writing, but the intent of the training is to expose the entire NYPD force to at least the basics of animal cruelty investigation and response.

lines 430-435  It is worth noting the inconsistency issues in US laws as well. As in Australia, in the US each state defines ‘animal’ and ‘animal cruelty’ differently and the laws vary widely in the range of penalties. Only a few laws address animal cruelty at the federal level (e.g. the Animal Welfare Act, the Endangered Species Act and the Marine Mammal Protection Act).

Author Response

There are some other additional possible causes of the enforcement gap that are worthy of mention in addition to the 4 listed. One is the complexity of animal cruelty offenses that require different areas of expertise and professional orientation. Animal cruelty can take many forms from simple neglect, ‘organized” abuse such as dogfighting and cockfighting, harm to pets in the context of domestic violence, gang-related activity, commercial exploitation such as in ‘puppy mills’ or extreme neglect seen in cases of animal hoarding. Responding to this range of offenses requires collaborative efforts that go beyond the expertise of traditional law enforcement or animal protection and often require the resources of mental health, social services, victim advocates, forensic veterinarians and others. A holistic approach should involve efforts for collaborative task forces and opportunities for cross-training in these different disciplines.

There are a lot of potential causes of the enforcement gap, and this comment is a fantastic example of that. You are exactly right that we will not see improvements in isolation but need collaboration with other professionals to improve this issue. In light of this comment, a section has been added in the conclusion stating that there are more causes of this gap that require further consideration and research (refer to line 786). We hope that more focussed and defined research will follow this introductory piece.

A second additional barrier is the lack of well-defined criteria for recognizing the physical and behavioral evidence associated with animal abuse and neglect. This has only recently begun to be addressed in the emerging fields of veterinary forensics and forensic applied animal behavior. Given growing public expectations of forensic evidence in any prosecution, the ‘CSI effect’ of the absence of such evidence can be a barrier to charging or fully prosecuting a cruelty case. Although these fields are growing in the US, UK and Brazil - there has not been much attention to these fields in Australia.

Again, another fantastic point that there is a disjoint between law, science and mental suffering/behavioural evidence. We have touched on this briefly at line 474 when talking about the doctrine of precedent and how resourcing can limit trial ‘test cases’. In Australia, most of the statutes include the definition of harm as “any form of pain, suffering, damage or distress”, so we do have the foundations to prosecute under mental suffering, however, the enforcement agencies seem hesitant to use behavioural evidence. This is definitely an area that requires future research.

A third additional contributor to the ‘gap’ is the absence of any centralized tracking of animal cruelty cases at the state/territorial or national level.  This has been a major issue in the US, with the FBI only beginning to attempt to track such cases in the last 3 years, with limited results so far.

Again, another good point. In Australia the data collection system in place is terrible. We have included a discussion on this on lines 774-782.  

lines 131-140 - In addition to the stated obstacles to prosecution of livestock cruelty cases, it may be worth mentioning that at least in the US there are specific livestock industry efforts to  directly block the investigation of such industries through hotly debated legislative efforts to pass “Ag-gag” laws that outlaw or severely restrict such investigations. I do not know if there have been such efforts in Australia, but they issue is worth mentioning.

Thank you for this suggestion. We have included a discussion on line 229-234 regarding ag-gag laws in Australia.

lines 369-374  It is worth mentioning any public campaigns that serve to better educate the public to recognize and report suspected animal cruelty, including specific high profile crimes such as dogfighting. As noted, a key need is also to provide the public with up-to-date information on where to report suspected animal cruelty, In the US a good resource for finding appropriate responders is provided at nationallinkcoalition.org.

Thank you for this suggestion, we have included a statement and reference to a campaign that RSPCA SA has run recently to better educate the public (refer to line 545).

line 390 There is slight error here. The collaborative effort between the ASPCA and the New York Police Department involves 33,000 police officers. I believe the 14,000 number for Coulter’s study represents the number trained as of her writing, but the intent of the training is to expose the entire NYPD force to at least the basics of animal cruelty investigation and response.

Thank you for picking this up. Due to the confusion around the figures and having a citation, the figures have been removed from the review (refer to line 565).

lines 430-435  It is worth noting the inconsistency issues in US laws as well. As in Australia, in the US each state defines ‘animal’ and ‘animal cruelty’ differently and the laws vary widely in the range of penalties. Only a few laws address animal cruelty at the federal level (e.g. the Animal Welfare Act, the Endangered Species Act and the Marine Mammal Protection Act).

Thank you for this suggestion, however, for the purposes of this review we would like to keep it focused on the Australian animal welfare enforcement given this is the jurisdiction where we have personal experience. Also, we would not wish to mislead readers by not presenting all the relevant US information. 

Round 2

Reviewer 2 Report

The second version of the paper is much better than the first and the additions from the authors have assisted in clarifying previous opaque points and assisting in defining the enforcement gap. There are valid arguments in looking at each of the stages and assessing if this contributes to the enforcement gap (reporting cruelty, investigating authority, law itself sentencing.

The authors need to consider whether they have devoted different levels of priority to each of these four areas: the law itself merits 2 pages, reporting cruelty 2 pages, the investigating authority 5 pages and sentencing 2 pages.  This may reflect that the piece is about the challenges of the investigating authority in the enforcement gap rather than the other three areas but if this is the author's conclusion (although there is no evidence to suggest this) this needs to be clearer in the conclusion otherwise the reader would be left with this assumption. 

The only comments are as follows

l 202 people in the livestock industry are also less likely to report cruelty as there are specific laws governing the farming regimes and these make it impossible to convict someone of animal cruelty if that method of farming is allowed eg there is evidence to show battery cage laying hen farming has welfare and cruelty problems but this is permitted in Australia and a farmer cannot be convicted of animal cruelty if they farm laying hens in a battery cage  - there is good juris prudence to show a court will not allow this eg https://www.casemine.com/judgement/uk/5a8ff7b360d03e7f57eb1532

l 413 whilst the issue of resourcing is now caveated that it refers to the literature it may be beneficial to source other literatures that show resourcing is not an issue for the RSPCA in deciding whether to prosecute eg https://www.rspca.org.uk/whatwedo/endcruelty/prosecution 

https://www.alaw.org.uk/2017/04/rspca-releases-latest-prosecutions-annual-report/ 

RSPCA 2017 Prosecutions Review "the Prosecutions Policy sets out the basis on which the RSPCA prosecutes and the Guidance followed during our reivew and decision making process". 

RSPCA Prosecution Policy 2017 file:///C:/Users/006983/Downloads/RSPCA_Prosecution_Policy_2017.pdf 

l 354 it is not clear why there is this reference to RSPCA UK as most of the paper is concerned with Australia - RSPCA UK has not received "significant criticism" for its campaigning - the Government has stated that they are quite happy for the RSPCA to campaign and prosecute so either this section should be removed or should be put into context by quoting the Wooler report.

Author Response

The authors need to consider whether they have devoted different levels of priority to each of these four areas: the law itself merits 2 pages, reporting cruelty 2 pages, the investigating authority 5 pages and sentencing 2 pages.  This may reflect that the piece is about the challenges of the investigating authority in the enforcement gap rather than the other three areas but if this is the author's conclusion (although there is no evidence to suggest this) this needs to be clearer in the conclusion otherwise the reader would be left with this assumption. 

The length of each section is more representative of the available literature rather than the priority of the sections. We think all sections warrant the same considerations, especially since we do not know the level of contribution they each make to the enforcement gap concept. It just happens that literature on the enforcement authorities more prevalent than the other three areas. This is likely due to the confidential nature of animal law. Also, because the RSPCA as an enforcement authority is rather transparent, which allows for more resources and literature on this area. We hope this clears it up.

The only comments are as follows

l 202 people in the livestock industry are also less likely to report cruelty as there are specific laws governing the farming regimes and these make it impossible to convict someone of animal cruelty if that method of farming is allowed eg there is evidence to show battery cage laying hen farming has welfare and cruelty problems but this is permitted in Australia and a farmer cannot be convicted of animal cruelty if they farm laying hens in a battery cage  - there is good juris prudence to show a court will not allow this eg https://www.casemine.com/judgement/uk/5a8ff7b360d03e7f57eb1532

This is a great point; however, we think the meaning may have been misconstrued. The paper we referenced (Taylor, N & Signal, T.D. “Community Demographics and the Propensity to Report Animal Cruelty”) discusses the propensity to report any acts of cruelty towards any animal by livestock workers, it is not specific to livestock species and therefore does not relate to the standards governing the livestock industry (ie. regulations and codes). We have attempted to clarify this point by altering the wording (refer to line 201).

l 413 whilst the issue of resourcing is now caveated that it refers to the literature it may be beneficial to source other literatures that show resourcing is not an issue for the RSPCA in deciding whether to prosecute eg https://www.rspca.org.uk/whatwedo/endcruelty/prosecution 

https://www.alaw.org.uk/2017/04/rspca-releases-latest-prosecutions-annual-report/ 

RSPCA 2017 Prosecutions Review "the Prosecutions Policy sets out the basis on which the RSPCA prosecutes and the Guidance followed during our reivew and decision making process". 

RSPCA Prosecution Policy 2017 file:///C:/Users/006983/Downloads/RSPCA_Prosecution_Policy_2017.pdf 

We thank you for providing us with these resources. However, given the resourcing section (section 4.3) of the paper is focused on Australia, we believe these resources are not appropriate as they are UK based. Your later comment also alludes to this focus on Australia, so after reflection we have decided not to include these resources.

l 354 it is not clear why there is this reference to RSPCA UK as most of the paper is concerned with Australia - RSPCA UK has not received "significant criticism" for its campaigning - the Government has stated that they are quite happy for the RSPCA to campaign and prosecute so either this section should be removed or should be put into context by quoting the Wooler report.

We have now removed this comment regarding RSPCA UK.